# Auto-Regulation of Transcription and Translation: Oscillations, Excitability and Intermittency

**DOI:** 10.3390/biom11111566

**Published:** 2021-10-22

**Authors:** Philip J. Murray, Eleonore Ocana, Hedda A. Meijer, Jacqueline Kim Dale

**Affiliations:** 1School of Science and Engineering, University of Dundee, Dundee DD1 4HN, UK; e.ocana@dundee.ac.uk; 2School of Life Sciences, University of Dundee, Dundee DD1 5EH, UK; HMeijer001@dundee.ac.uk (H.A.M.); j.k.dale@dundee.ac.uk (J.K.D.)

**Keywords:** post transcriptional regulation, translation, mathematical model, excitable, oscillatory, threshold, intermittency

## Abstract

Several members of the Hes/Her family, conserved targets of the Notch signalling pathway, encode transcriptional repressors that dimerise, bind DNA and self-repress. Such autoinhibition of transcription can yield homeostasis and, in the presence of delays that account for processes such as transcription, splicing and transport, oscillations. Whilst previous models of autoinhibition of transcription have tended to treat processes such as translation as being unregulated (and hence linear), here we develop and explore a mathematical model that considers autoinhibition of transcription together with nonlinear regulation of translation. It is demonstrated that such a model can yield, in the absence of delays, nonlinear dynamical behaviours such as excitability, homeostasis, oscillations and intermittency. These results indicate that regulation of translation as well as transcription allows for a much richer range of behaviours than is possible with autoregulation of transcription alone. A number of experiments are suggested that would that allow for the signature of autoregulation of translation as well as transcription to be experimentally detected in a Notch signalling system.

## 1. Introduction

The Notch signalling pathway plays a crucial role in many different developmental contexts (e.g., neural, endocrine, cardiovascular), is involved in the function and maintenance of adult tissue (e.g., [1]) and is known to be aberrant in many cancers (e.g., [2]). Notch signalling is associated with diverse pattern formation phenomena such as lateral inhibition, lateral induction and oscillator synchronisation [3].

Notch signalling occurs when a Notch ligand (i.e., Serrate or Delta) interacts with a Notch receptor. Upon activation of the Notch receptor, the Notch intracellular domain is cleaved and transported to the nucleus where it activates target gene expression [4]. Interaction with the Notch receptor can either be via a neighbouring cell (trans) or within a single cell (cis). Amongst the targets of Notch signalling are members of the basic helix-loop-helix family of transcription repressors (e.g., Hes7, Hes1, Her7). When in dimerised form, these proteins inhibit their own transcription and therefore provide a negative feedback loop in the Notch signalling pathway [5].

Negative feedback loops play an essential role in biological oscillations [6]. The Goodwin [7] model considers an *N*-component cascade that is typically formulated such that the Nth component in the cascade closes the feedback loop by nonlinearly inhibiting the first component. Given at least three members in the cascade and sufficient nonlinearity in the inhibition, oscillatory solutions can be obtained. However, in the presence of time delays that represent the time required to, for example, transcribe, splice, transport and translate mRNAs, oscillations are possible in a two variable cascade [8,9]. It has been shown that transport processes, explicitly modelled using reaction diffusion theory, are themselves sufficient to give rise to oscillatory solutions [10,11]. A common feature of the models described above is that processes such as mRNA degradation and translation are assumed to be linear. However, in many situations such processes are regulated and therefore unlikely to be linear.

Positive feedback loops have also been identified in many biological oscillators. For example, in models of the cell cycle positive feedback loops give rise to hysteresis that results in cell cycle phase transitions [12]. In neurons positive feedback between the membrane potential and the conductance of particular ion channels gives rise to membrane excitability and oscillations [13]. It has been suggested that one desirable feature of positive-feedback in biological systems is tunability: the ability of an oscillator to exhibit a large range in frequency whilst maintaining approximately constant amplitude [14].

Levels of gene products are regulated by processes such as transcriptional activation, post transcriptional regulation (from splicing to RNA stability) and protein degradation. mRNA sequences contain untranslated regions (UTRs) that flank the open reading frame at the 5′ and 3′ ends. UTRs often contain conserved sequence elements that facilitate control of translational efficiency and mRNA stability through the binding of specific regulatory factors. Regulators at the post transcriptional level include RNA binding proteins (RNA-BPs), microRNAs (miRNAs) and long-noncoding RNAs (lncRNAs). The 5′UTR is mostly involved in regulating translation efficiency with elements such as upstream open reading frames and secondary structures inhibiting translation. Other sequences such as 5′ terminal oligopyrimidine sequences and internal ribosome entry sites may enable translation under conditions where most other mRNAs are translationally inactive. The 3′UTRs often contain several binding sites for miRNAs and/or RNA-BPs that mostly inhibit translation and/or target the mRNA for degradation.

In mouse neural progenitor cells it has been identified that: (i) the miRNA mir-9 is required for Hes1 oscillations, (ii) Hes1, a Notch target gene, represses miR-9 transcription; and (iii) miR-9 decreases Hes1 mRNA stability, resulting in a double negative feedback loop [15]. The disproportionate effect of mir-9 overexpression on Hes1 protein levels has also led to the suggestion of an additional effect of mir-9 on mRNA translation [15]. The role of mir-9 has been explored in a delay differential equation model of the Hes1 [16] system. In the segmentation clock, a Notch-dependent molecular oscillator that drives oscillatory expression of segmentation clock genes in a periodic fashion during early embryogenesis, it has been established that the 3′UTR is critical for the correct expression of several segmentation clock genes and some interacting miRNAs have been identified [17]. Thus there is strong evidence for post-transcriptional regulation in two distinct developmental systems in which Notch signalling plays a crucial role.

The aim of this study is to investigate a potential role for translational regulation in a model of a transcriptional repressor. The layout is as follows: in Section 2 we introduce a mathematical model; in Section 3.1 we explore numerical results and in Section 3.2 identify experimentally testable predictions from the model; and, finally, in Section 4 we conclude with a discussion.

## 2. Materials and Methods

### 2.1. Model Development

Let M=M(t), P=P(t) and X=X(t) represent the number of molecules of mRNA, protein and X, respectively, in a cell at time *t*. It is assumed that transcription is repressed by protein dimers such that the transcription rate is given by
$$\frac{k_{1}}{1+{\left(\frac{P}{P_{0}}\right)}^{2}},$$
where $k_{1}$ represents the maximal production rate of mRNA and $P_{0}$ the number of molecules of protein at which mRNA transcription is half maximal. Note that this functional form is equivalent to that considered by Lewis [9]. It is additionally assumed that translation of mRNA occurs at background translation rate, $k_{3}$, and at a nonlinear rate, $k_{4}$, that is inhibited by *X*, i.e. translation occurs at rate
$$k_{3}+\frac{k_{4}}{1+\frac{X}{X_{0}}},$$
where $X_{0}$ is the number of molecules of *X* at which *X*-dependent translation is half-maximal. Finally, *X* is assumed to be under the same transcriptional regulation as *M*.

The total rates of change with respect to time are assumed to be given by the ordinary differential equations
(1)$$\begin{array}{cc} \frac{dM}{dt} & =\frac{k_{1}}{1+{\left(\frac{P}{P_{0}}\right)}^{2}}-k_{2}M,\\ \frac{dP}{dt} & =M\left(k_{3}+\frac{k_{4}}{1+\frac{X}{X_{0}}}\right)-k_{5}P,\\ \frac{dX}{dt} & =\frac{k_{6}}{1+{\left(\frac{P}{P_{0}}\right)}^{2}}-k_{7}X, \end{array}$$
where $k_{2}$ is the mRNA degradation rate, $k_{5}$ is the protein degradation rate, $k_{6}$ is the production rate of *X* and $k_{7}$ is the degradation rate of *X*.

In this study we make a quasi-steady-state approximation for *X*. This assumption could be realised, for example, in a situation where the production and degradation rates of *X* are relatively large. Hence
$$X=\frac{\alpha }{1+{\left(\frac{P}{P_{0}}\right)}^{2}},$$
where
$$\alpha =\frac{k_{6}}{k_{7}}.$$
Substitution for *X* into Equations (Equation 1) yields
(2)$$\begin{array}{cc} \frac{dM}{dt} & =\frac{k_{1}}{1+{\left(\frac{P}{P_{0}}\right)}^{2}}-k_{2}M,\\ \frac{dP}{dt} & =M\left(k_{3}+\frac{k_{4}}{1+\frac{\alpha }{X_{0}}\frac{1}{\left(1+{\left(\frac{P}{P_{0}}\right)}^{2}\right)}}\right)-k_{5}P, \end{array}$$
with initial conditions given by
$$M\left(0\right)=M_{0},\;P\left(0\right)=P_{0}.$$
Equations (Equation 2) were solved numerically using the Matlab ODE solver ode15s.

### 2.2. Parameters

Parameter values have been chosen based on a model of the Her oscillator [9]. Transcription and protein degradation are assumed to occur on a time scale of minutes but, notably, the mRNA degradation rate is smaller than that considered by Lewis [9]. See Table 1.

### 2.3. Bifurcation Analysis

Bifurcation diagrams were produced using the software coco [18].

### 2.4. Stochastic Model

A stochastic model is obtained upon making the assumption of additive white noise in both mRNA and protein dynamics. Thus Equation (Equation 2) generalise to
(3)$$\begin{array}{cc} \frac{dM}{dt} & =\frac{k_{1}}{1+{\left(\frac{P}{P_{0}}\right)}^{2}}-k_{2}M+\sigma_{M}\xi_{M}\left(t\right),\\ \frac{dP}{dt} & =M\left(k_{3}+\frac{k_{4}}{1+\frac{\alpha }{X_{0}}\frac{1}{\left(1+{\left(\frac{P}{P_{0}}\right)}^{2}\right)}}\right)-k_{5}P+\sigma_{P}\xi_{P}\left(t\right), \end{array}$$
where $\sigma_{M}$ and $\sigma_{P}$ represent noise strengths and $\xi_{M}\left(t\right)$ and $\xi_{P}\left(t\right)$ Gaussian noise. Equation (Equation 3) were solved numerically using the Euler-Maruyama method.

## 3. Results

### 3.1. Exploring Model Behaviour

A model is considered that accounts for the interactions between three components: mRNA, M, protein, P, and a translation inhibitor, X. It is assumed that: (i) transcription of mRNA is inhibited by protein dimers [9]; (ii) X is under the same transcriptional control as M; (iii) X inhibits translation (see Figure 1a); and (iv) all species undergo linear degradation [9]. In Section 2.1, a set of ordinary differential equations is derived that describes the rates of change of the different molecular species.

The inhibition of translation by X introduces a double negative feedback loop (P represses production of X which represses translation of P), i.e., a positive feedback loop. Hence the translation rate is an increasing sigmoidal function of *P*. In contrast, the transcription rate is a decreasing function of *P* (see Figure 1b). For large *P*, the translation rate tends to its maximal value $k_{3}+k_{4}$ whilst when *P* is small the translation rate is reduced to $k_{3}$. In this study we will consider the case where: (i) $k_{4}\gg k_{3}$ (i.e., X can potentially have a large effect of the effective translation rate); (ii) *X* is in quasi-equilibrium; and (iii) mRNA is more stable than protein ($k_{2}<k_{5}$).

In the limit of a low baseline transcription rate (small $k_{1}$), the model has a unique stable steady state. For a representative initial condition, both *M* and *P* evolve to the steady state (see Figure 2a). It is instructive to plot the solution in the phase plane (see Figure 2b); here *M* and *P* are plotted in a plane such that the time series solutions in Figure 2a are represented by the solid magenta line in Figure 2b. The *M* and *P* nullclines represent points in the plane where the time derivatives are zero (e.g., the *P* nullcline represents the steady state levels of protein for a given ‘clamped’ mRNA level, and vice-versa). Note that the *P* nullcline has, for the chosen parameters, a local maximum and minimum, but that the *M* nullcline is monotonically decreasing. The solution trajectory ultimately tends to the intersection of the nullclines. In the limit of low transcription rate, the system behaves like a standard model of auto-inhibition of transcription (e.g., a two variable Goodwin model).

Upon increasing the background transcription rate, $k_{1}$, the behaviour of the solution is qualitatively different: a small but finite perturbation can result in a pulse of protein before the solution returns to equilibrium. For a small perturbation about the steady state (see Figure 3a,b), the solution quickly returns to the steady state and levels of protein do not change markedly. However, when the perturbation is large enough (see Figure 3c,d), a transient pulse of protein is produced and the solution takes a relatively long time to return to steady state. These numerical results are indicative of a thresholding phenomenon whereby sufficiently large stimuli (in this case provided via the initial conditions) result in a large excursion in the phase plane (in this case a transient pulse of protein). By considering different initial perturbations of both M and P, we identify threshold perturbations that trigger the transient protein response (see Figure 3e,f).

The large amplitude trajectory presented in Figure 3b,d, annotated by ABCDE, can be explained as follows: at A levels of mRNA have been increased such that sufficient protein can be generated to switch the translation rate from low to high. As $k_{4}\gg k_{3}$, a large amount of protein is produced (the solution reaches B). However, as transcription is inhibited by high protein levels, mRNA levels deplete and there is a reduction in protein (B to C). As protein levels decrease, the translation rate ultimately switches back to the lower rate (C to D) and transcription is again active. Finally, levels of mRNA increase and the solution returns to the steady state (D to E).

Further increasing the transcription rate, $k_{1}$, results in the emergence of two large amplitude limit cycle solutions (one stable and one unstable, see Figure 4). However, the steady state remains linearly stable, with trajectories that are initialised close to it tending towards it (see Figure 4a,b). In contrast, solutions that are initialised sufficiently far from the stable steady state instead find the stable limit cycle (see Figure 4c,d). Hence for a particular range of the transcription rate $k_{1}$, the model is bistable.

It has been shown that isolated zebrafish presomitic mesoderm (PSM) cells exhibit intermittent oscillations whereby successive cycles are followed by periods of non-oscillatory behaviour [19]. Upon introducing a term representing white noise in the mRNA and protein dynamics, we obtain a stochastic differential equation model (see Section 2.4). With appropriately chosen noise strength, the solutions of the stochastic model toggle between stable steady states and the stable limit cycle (see Figure 4e). Hence the proposed model possesses a parameter regime in which there are intermittent oscillations.

Returning to the deterministic model, further increase in $k_{1}$ results in a subcritical Hopf bifurcation: the stable steady state and unstable limit cycle are lost and solutions tend to the stable limit cycle (see Figure 5). The stable limit cycle solutions can be characterised by the trajectory ABCDA (see Figure 5). At A, levels of M are high and P are low. However, translation of the mRNA pool generates sufficient P to activate the translation switch, levels of P increase and the system moves to B. Higher levels of P then result in inhibition of transcription and the mRNA pool depletes (B to C). On the segment CD levels of protein decrease and the translation switch is deactivated. At D the transcription rate increases again as levels of P are low and the system moves back towards A. Hence in this model the coordinated switching on/off of both transcription and translation is sufficient to give rise to limit cycle solutions.

The numerical results presented in Figure 2, Figure 3 and Figure 4 are summarised in the bifurcation diagram in Figure 6: for small $k_{1}$ there is a stable steady state; for an intermediate range of (approximately $1.5<k_{1}<1.6)$, there is a stable steady state, an unstable limit cycle and a stable limit cycle. At $k_{1}∼1.6$ there is a subcritical Hopf bifurcation at which the stable steady state becomes unstable. At $k_{1}∼15.8$ there is a supercritical Hopf bifurcation; there is the emergence of small amplitude stable limit cycles. See Figure 5 for a plot of the oscillator period against $k_{1}$.

### 3.2. Experimentally Testable Predictions

#### 3.2.1. Inhibition of Transcription via a Notch Signalling Inhibitor

γ secretase inhibitors (e.g., LY411575) provide a dose-dependent means of inhibiting Notch signalling via inhibiting the release of the Notch intracellular domain (NICD) from the Notch receptor (e.g., [20]). Although NICD is not explicitly accounted for in the proposed model, we reasoned that the parameter in the model most strongly affected by NICD is the basal transcription rate $k_{1}$. Hence below we consider reduction in $k_{1}$ as a proxy for dose-dependent γ secretase inhibitor treatment.

Upon transcriptional block ($k_{1}=0$), the time course of protein levels has a distinct and testable signature. To illustrate this we consider a representative sample of points from the limit cycle solution (see Figure 7a) as initial conditions at which to simulate transcriptional block (i.e., set $k_{1}=0$). Independently of the position in the cycle at which transcription is blocked, mRNA levels decrease exponentially to zero (see Figure 7b). However, in contrast to mRNA, the time course of protein levels is strongly dependent on the position in the cycle at which the perturbation is applied. When protein levels are low, protein also decays exponentially. However, when protein levels are high, the decay is bimodal (see Figure 7). Thus measurement of the decay kinetics of protein, P, upon complete transcriptional inhibition has a cycle-dependent signature. This effect could be quantified, for example, using a real time reporter of protein (e.g., Hes7-Achilles [21]).

Partial inhibition of transcription in the oscillatory regime can result, counterintuitively, in an increase in detectable mRNA levels. Consider the case where a population of asynchronous cellular oscillators is subject to Western blot and qPCR in order to quantify levels of P and M, respectively. In the context of the proposed model, the measured quantities can be represented by the time-average of the oscillatory signal, i.e.,
(4)$$\begin{array}{c} \bar{M}=\frac{1}{T}\int_{t-T}^{T}M\left(t\right)dt\;\;\bar{P}=\frac{1}{T}\int_{t-T}^{T}P\left(t\right)dt, \end{array}$$
where *T* is the oscillator period. We find that levels of mRNA and protein broadly decrease, as expected, as the transcription rate $k_{1}$ decreases and ultimately oscillations are lost (see Figure 8). However, at the bifurcation value of $k_{1}$, where oscillations are lost, the system becomes excitable, with the steady state having relatively high levels of mRNA. Hence, in principle, increased levels of mRNA can be detected upon decreasing the transcription rate $k_{1}$ (see Figure 8b) as the steady state in the excitable regime has a higher level of mRNA than that obtained by time-averaging the oscillations.

#### 3.2.2. Treatment with a Translational Inhibitor

We next investigated how the model behaves upon inhibition of translation. Translation could, in the context of the model, be inhibited in two specific ways: reduction in the background translate rate, $k_{3}$, or the X-dependent translation rate, $k_{4}$. Cycloheximide (CHX), for example, is a general inhibitor of eukaryotic translation elongation which can be used in a dose dependent manner to slow or stop translation. Moreover, the translation rates of individual mRNAs are dependent on specific sequence elements in the mRNA sequence, mostly in the 5′ and 3′ untranslated regions, and potentially the factors that can interact with these (represented by X). Mutagenesis of sequence elements or removal/inhibition or overexpression of the interacting factors can affect translation rates of specific mRNAs.

To model specific translation inhibition, we consider reduction in the parameter $k_{4}$. Upon complete inhibition of X-dependent translation ($k_{4}=0$), levels of protein decrease to a low steady state whilst levels of mRNA tend to a high steady state as there is less protein to inhibit transcription (see Figure 9a,b). Upon partial inhibition of mRNA translation rate, levels of protein decrease whilst levels of mRNA increase (see Figure 9c–f). Moreover, the period of the oscillation increases as the parameter $k_{4}$ decreases (see Figure 9d). These numerical results indicate that perturbation of translation rates ought to result in detectable changes to levels of mRNA and protein as well as the clock period.

## 4. Discussion

In homeostatic systems the presence of negative feedback loops can lead to the product(s) of a pathway repressing further transcription, thus resulting in a stable steady state. In oscillatory systems negative feedback loops in the presence of delays can give rise to oscillatory gene expression patterns.

Post-transcriptional modifications encompass a wide range of processes that involve the processing, translation and stability of mRNAs. Whilst they have been shown to play a crucial role in the regulation of the circadian oscillator, far less is known about the role of post transcriptional modifications in the Notch signalling pathway. Indeed, most mathematical models of mRNA degradation and translation in the Notch signalling pathway assume that such processes are linear and unregulated. In this study we consider a feedback loop that ultimately results in the translation rate of a protein increasingly sigmoidally with protein levels (i.e., a positive feedback loop). Our results show that, in principle, such post-transcriptional regulation can give rise to a system that is reminiscent of the FitzHugh-Nagumo [22,23] model of nerve excitation [13]. The model possesses a rich family of dynamical behaviours that a cell could in principle use to regulate aspects of its behaviour.

For intermediate background transcription rates, the model is excitable. In this regime small perturbations to the steady state result in the system quickly returning to steady state. However, a sufficiently large perturbation results in a transient pulse of protein levels before the system eventually returns to the stable steady state. Notably, it has previously been suggested that the segmentation clock oscillator is excitable [24]. Whilst it has been shown that a Yap/Hippo/NICD axis regulates the segmentation clock, the precise details of the molecular circuitry that underlies such excitability have not been determined.

For larger transcription rates the model is bistable: there is a stable steady state and a stable limit cycle. Depending on the initial conditions that are chosen, model solutions tend to one of the identified attractors. Moreover, upon the inclusion of additive white noise, the model solutions can switch between the two stable states, behaviour that is reminiscent of observations of intermittent oscillations in isolated zebrafish PSM cells [19]. Notably, in the previous study the authors considered a Stuart Landau model, which has a supercritical Hopf bifurcation. In the proposed model, intermittent oscillations can be observed close to the onset of an unstable limit cycle (and hence a subcritical Hopf bifurcation). This allows for the possibility of noise to switch the system between large amplitude limit cycle solutions and a stable steady state.

Phenomenology in the model is consistent with some experimental observations of the zebrafish segmentation clock. When Notch signalling is inhibited, via DAPT treatment [25], the oscillator period increases. Moreover, when levels of the Notch ligand DeltaD are increased, the clock period decreases [26]. Together, these results suggest that in the zebrafish segmentation clock, levels of Notch signalling are anticorrelated with the clock period. These results are consistent with the predicted dependence of the oscillator period on the transcription rate $k_{1}$ (see Figure 8). However, it has been shown in mouse and chick embryos that pharmacological perturbation that resulted in increased levels of NICD are correlated with a longer clock period [20]. A more detailed exploration of the model is required to investigate if the computed dependence of oscillator period on the parameter $k_{1}$ is universal or specific to the parameter values chosen.

A further prediction of the model is that a reduction in the transcription rate could lead to higher levels of mRNA. In a system of uncoupled cellular oscillators, where one samples the average level of mRNA using a technique such as qPCR, the steady state level of mRNA measured close to the point where oscillations are lost will be higher than the average over the oscillatory cycle. Hence one could observe that, at the population average level, transcriptional inhibition counterintuitively results in an increase in mRNA levels.

In this study a reference set of parameter values has been chosen based upon a previous model of delayed negative feedback oscillations. So as to allow focus of the interaction between nonlinearities in translation and transcription, we have not explicitly accounted for delays that represent, for example, splicing and transport. Moreover, we have focussed here only on the sensitivity of model solutions to the parameter $k_{1}$ as it is experimentally accessible. In a future study we will systematically explore model behaviour in a more general setting.

The quasi-steady state approximation for the variable *X* allows for the model analysis to be simplified and therefore for parallels with excitable medium theory to be explored. Whilst the quasi-steady state approximation could be realised, for example, if *X* were relatively unstable and produced at a relatively large rate, further experimental work is required to firstly identify molecular regulators of post-transcriptional regulation and then to quantify their kinetics. In mouse neural progenitors [15] it has been identified that transcription of the miRNA mir-9 is coregulated with transcription of the Hes1 gene. However, in this system mir-9 is found to be relatively stable and accumulates slowly over many cycles of the Hes1 oscillators.

It is a well-established result that combining positive and negative loops can give rise to nonlinear behaviours such as excitability. Here the major novelty is the application of this idea to the Notch signalling pathway. However, in the derivation of the proposed model we hypothesised that mRNA translation is negatively regulated by an unidentified molecule X. Whilst there are prospective molecular candidates that could satisfy the assumptions made in the model development, further work is required to experimentally determine the molecular players.

## Figures and Tables

**Figure 1 biomolecules-11-01566-f001:**
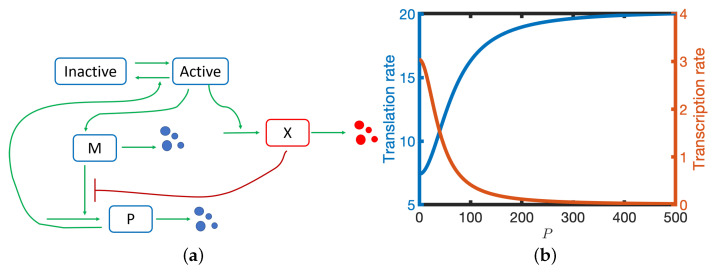
(**a**) A schematic diagram of the model. M—mRNA, P—protein, X—translational inhibitor. (**b**) The translation (blue line) and transcription (red line) rates are plotted against *P*. See Table 1 for parameter values.

**Figure 2 biomolecules-11-01566-f002:**
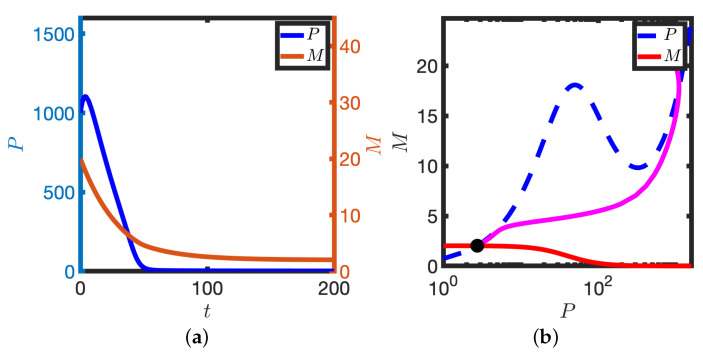
Homeostasis in the case of low transcription rate. (**a**) *P* and *M* are plotted against time, *t*. (**b**) The corresponding solution (magenta line) is plotted in the PM phase plane. P nullcline (dashed blue line), M nullcline (solid red line), Steady state (black dot). Equations (Equation 2) were solved numerically. $k_{1}=0.234$. Other parameter values in Table 1.

**Figure 3 biomolecules-11-01566-f003:**
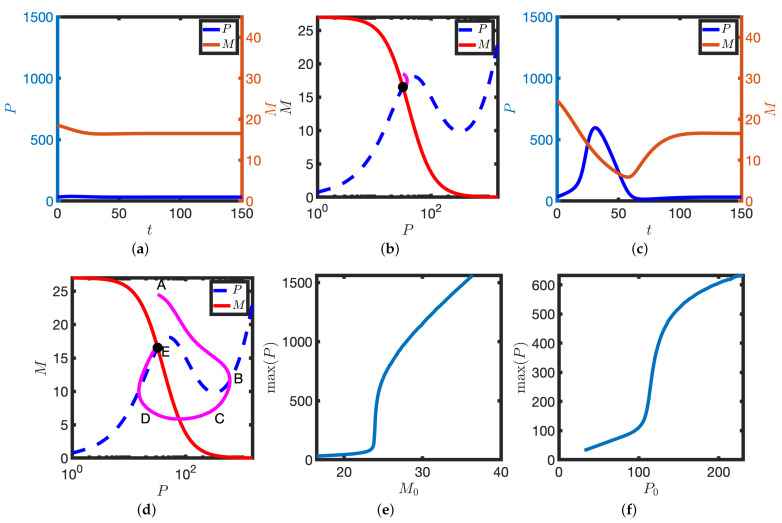
Excitable solutions in the case of an intermediate transcription rate. (**a**,**b**) Solution dynamics upon making a small initial perturbation about the steady state. *P* and *M* are plotted against time, *t*, (**a**) and in the phase plane (**b**). (**c**,**d**) Solution dynamics upon making a large initial perturbation about the steady state. Other details as in (**a**,**b**). (**b**,**d**) Magenta lines represent solutions in the PM phase plane. P nullclines (dashed blue lines), M nullclines (solid red lines), steady states (black dots). (**e**) max(P) is plotted against the initial number of mRNA molecules, $M_{0}$. (**f**) max(P) is plotted against the initial number of protein molecules, $P_{0}$. $k_{1}=0.81$. Equation (Equation 2) were solved numerically. Other parameter values as in Table 1.

**Figure 4 biomolecules-11-01566-f004:**
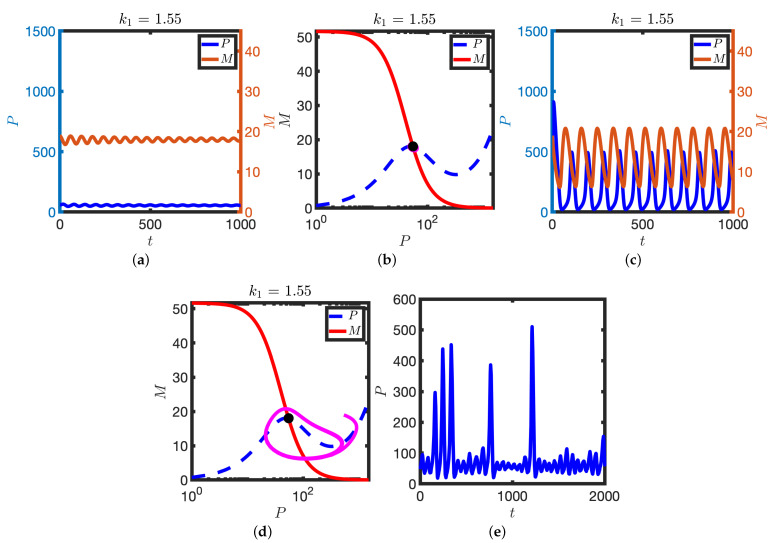
The emergence of a stable limit cycle upon further increase in the parameter $k_{1}$. (**a**,**b**) Solution dynamics following a small perturbation about the steady state. $M_{0}=54$, $P_{0}=19$. (**a**) *P* and *M* are plotted against time, *t*. (**b**) Corresponding solutions (magenta lines) are plotted in the PM phase plane. P nullclines (dashed blue lines), M nullclines (solid red lines), steady states (black dots). (**c**,**d**) A larger perturbation about the steady state results in a stable limit cycle. $M_{0}=540$, $P_{0}=19$. Other details as in (**a**,**b**). Equation (Equation 2) were solved numerically. (**e**) Intermittent oscillations: noise allows for stochastic switching between stable limit cycle and the stable steady state. *P* is plotted against time, *t*. Equation (Equation 3) were solved numerically. $k_{1}=1.55$. Other parameters as in Table 1.

**Figure 5 biomolecules-11-01566-f005:**
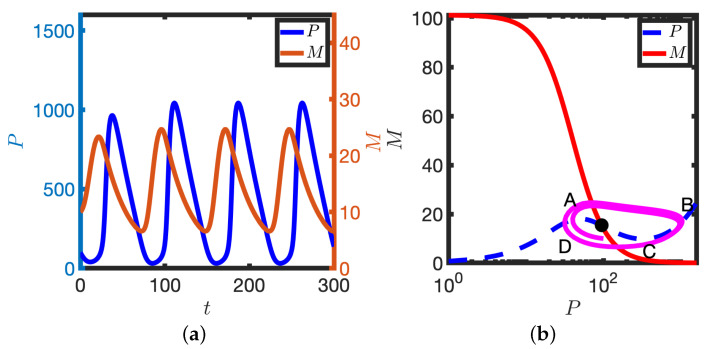
Oscillatory solutions of Equation (Equation 2). (**a**) *P* and *M* are plotted against time, *t*. (**b**) Corresponding solution (magenta line) is plotted in the PM phase plane. P nullcline (dashed blue line), M nullcline (solid red line), steady states (black dot). See Table 1 for parameter values.

**Figure 6 biomolecules-11-01566-f006:**
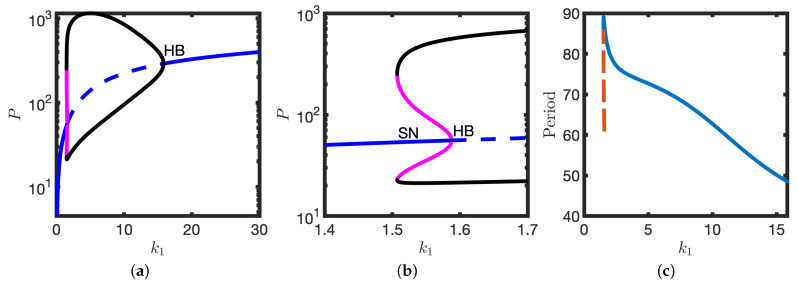
Bifurcation structure of Equation (Equation 2) upon varying the parameter $k_{1}$. (**a**) Number of protein molecules is plotted against $k_{1}$. Extrema of the stable (black line) and unstable (magenta line) limit cycles. Stable steady state (solid blue lines). Unstable steady state (dashed blue line). (**b**) Inset for (**a**). (**c**) Oscillator period is plotted against $k_{1}$. Stable limit cycle (solid line), unstable limit cycle (dashed line). Hopf bifurcation (HB), Saddle node bifurcation (SN). Other parameter values as in Table 1.

**Figure 7 biomolecules-11-01566-f007:**
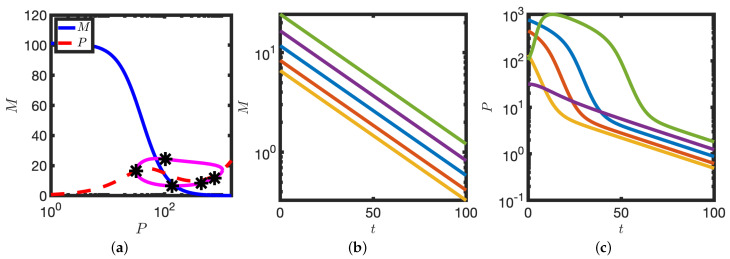
Protein decay is multimodal upon complete transcriptional inhibition. (**a**) A set of representative points (markers) is chosen on the limit cycle cycle solution. Other details as in Figure 5b. (**b**) mRNA levels are plotted against time. (**c**) Protein levels are plotted against time. Initial conditions given by each of the markers in (**a**). Equation (Equation 2) were solved numerically. See Table 1 for parameter values.

**Figure 8 biomolecules-11-01566-f008:**
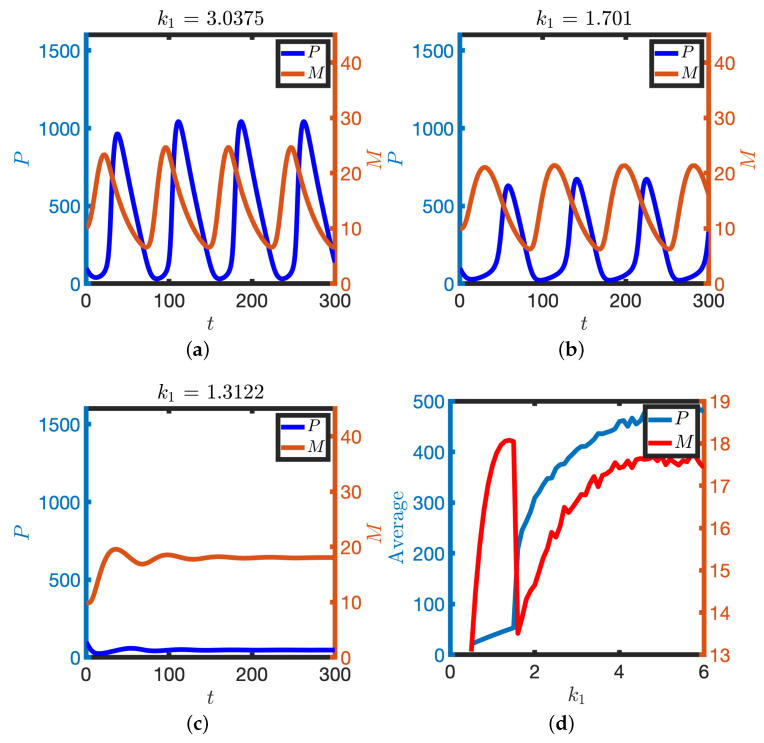
The effect of partial reduction of the transcription rate, $k_{1}$. (**a**–**c**) *M* and *P* are plotted against time for different transcription rates. (**d**) The detectable amount of M and P is plotted against $k_{1}$ (see Equation (Equation 4)). Equation (Equation 2) were solved numerically. Other parameter values as in Table 1.

**Figure 9 biomolecules-11-01566-f009:**
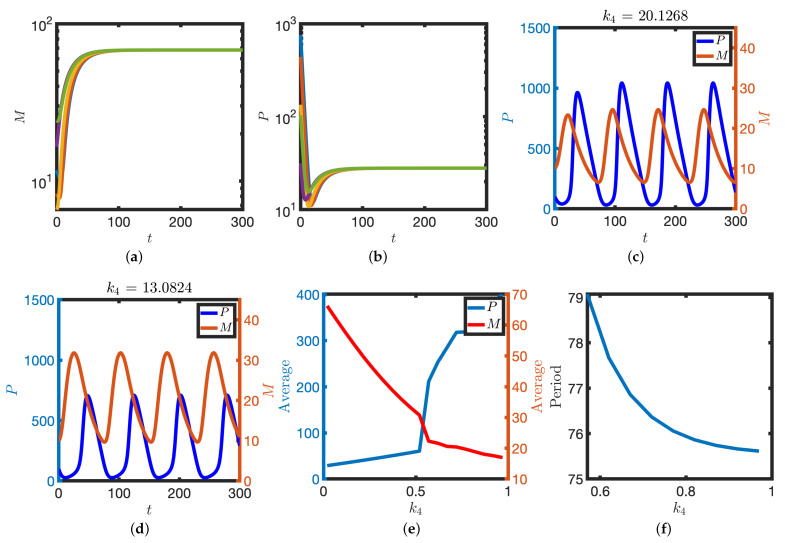
Perturbation of translation rates. (**a**,**b**) X-dependent translation is blocked ($k_{4}=0$). mRNA (**a**) and protein (**b**) levels are plotted against time. Initial conditions given by each of the markers in Figure 7a. (**c**–**f**) Partial block of translation rates. *M* and *P* are plotted against time for larger (**c**) and smaller (**d**) translation rates. (**e**) Time-averaged levels of M and P (Equation (Equation 4)) are plotted against the parameter $k_{4}$. (**f**) The period of the oscillator is plotted against the parameter $k_{4}$. Equation (Equation 2) were solved numerically. Other parameter values as in Table 1.

**Table 1 biomolecules-11-01566-t001:** A table of parameter values.

Parameter	Description	Value	Units
$k_{1}$	Maximal transcription rate	3.03	min${}^{-1}$
$k_{2}$	mRNA degradation rate	0.03	min${}^{-1}$
$k_{3}$	Background translation rate	0.12	min${}^{-1}$
$k_{4}$	X dependent translation rate	20.12	min${}^{-1}$
$k_{5}$	P degradation rate	0.30	min${}^{-1}$
α	Max. X	70.43	Nondim
$X_{0}$	Translation IC50	1	Nondim
$P_{0}$	Transcription IC50	40	Nondim
$\sigma_{M}$	Protein noise strength	1.0	min${}^{-1}$
$\sigma_{P}$	mRNA noise strength	0.12	min${}^{-1}$
$M_{0}$	mRNA initial condition	6	Nondim
$P_{0}$	protein initial condition	60	Nondim

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
