# Peer review of "Auto-Regulation of Transcription and Translation: Oscillations, Excitability and Intermittency"

_biomolecules, 2021, doi:10.3390/biom11111566_

Round 1
Reviewer 1 Report
Authors analyze a simple ODE model of a self-repressing gene inspired. by the Notch signaling pathway and Hes genes. They add a negative translational regulation by the protein. While they argue that translational regulation (either up-regulation or down-regulation happens, thy do not justify the choice of negative regulation by the protein that is being expressed, nor is there a justification of this type of regulation in Hes genes.
The model construction (given the assumptions) is correct. A surprising step in the analysis is the quasi-steady state approximation of translational repressor equation, but not the others. This is not justified in any way and it is not clear if the conclusions of the paper are valid for the 3D model in place of 2D model.
Numerical exploration of the reduced 2D model is clear. Results are not very surprising given that the model has a negative and a positive feedback loop. Existence of bistability between an equilibrium and a periodic orbit is nice, but sensitivity with respect to more than one parameter would be welcomed.
The section where model prediction for several experiments is very nice and well done. It may attract attention from experimental biologists.
Minor comments:
start of section 3.1 repeats start of section 2.1
Figure 6. is not clear: (b) is inset for (a) but there is no marking on (a). where this should go. Part (c) is not clear: what does vertical dashed line mean?
Author Response
Reviewer 1
- Authors analyze a simple ODE model of a self-repressing gene inspired by the Notch signaling pathway and Hes genes. They add a negative translational regulation by the protein. While they argue that translational regulation (either up-regulation or down-regulation happens, thy do not justify the choice of negative regulation by the protein that is being expressed, nor is there a justification of this type of regulation in Hes genes.
>>>In this study we explore a generic role for positive feedback in the Hes pathway (i.e. we are not committed to a particular cell type/environment). However, we do highlight that one particular instance of the interaction that we describe has been identified by Bonev et al who have shown that the micro mRNA mir-9 exhibits the functional role under study in this paper in the Hes1 system (i.e. the miRNA mir-9 is under the same transcriptional regulation as Hes1 but inhibits the translation of Hes1 protein). However, in the system considered by Bonev et al. it is believed that miRNA accumulates on a slow time scale. Thus the type of regulation that we have proposed can be justification in the regulation of Hes genes.
- A surprising step in the analysis is the quasi-steady state approximation of translational repressor equation, but not the others. This is not justified in any way and it is not clear if the conclusions of the paper are valid for the 3D model in place of 2D model.
>>>>The reviewer raises an interesting point and is correct that we have not explored the behaviour of the full 3D model. Here we have explored the limiting case of the 2D model and did not go further than this as the 2D model already resolves some of the issues with the consensus model of Notch signalling and allows for a number of experimentally testable predictions. We believe that further experimental work is firstly needed to determine molecular players in particular biological contexts. We have added a paragraph to the Discussion in which we highlight the reviewer’s point and introduced the quasi-steady state assumption with more detail and an example of how it could be realised. A more complete analysis of the model is being prepared for publication in a more theoretical journal.
- Numerical exploration of the reduced 2D model is clear. Results are not very surprising given that the model has a negative and a positive feedback loop.
>>>>The reviewer makes a fair point that the results are not that surprising (from a nonlinear systems perspective). However, we believe that the interesting and novel value here is the potential importance in the context of the Notch signalling pathway. This point has been noted in the discussion. 1
- Existence of bistability between an equilibrium and a periodic orbit is nice, but sensitivity with respect to more than one parameter would be welcomed.
>>>>Given the focus of this special issue, we have chosen to perform sensitivity analysis on one experimentally accessible parameter. We are currently preparing a more theoretical paper for publication in which we will perform an extensive parameter sensitivity analysis. This point has been noted in the discussion.
- start of section 3.1 repeats start of section 2.1
>>>>This issue has been addressed.
- Figure 6. is not clear: (b) is inset for (a) but there is no marking on (a). where this should go. Part (c) is not clear: what does vertical dashed line mean?
>>>(a,b) Yes, it is correct that (b) is the inset for (a). The position of the inset is not marked on (a) as the scale is too fine (k1 ∈ [1.4, 1.7] v k1 ∈ [0, 30]). (c) The dashed line represents the period of an unstable limit cycle. The caption has been modified to make this clearer.
Reviewer 2 Report
It is well known that Notch signalling pathway contributes to the autoinhibition of transcription which in turn can yield homeostasis. In the presence of delays that account for time needed for the processes such as transcription, splicing and transport, Notch signalling pathway can result in oscillations. Existing models of autoinhibition of transcription often naively assume translation as being unregulated (i.e. linear). The authors here propose a model with nonlinear regulation of translation. Without time delay, such a model can also yield rich dynamics including excitability, homeostasis, oscillations and intermittency. Moreover, the authors included a few suggestions of experiments for identifying the signature of autoregulation of translation as well as transcription in a Notch signalling system. The paper is mostly well written. However, the model was presented with minimum details and justifications. This can be improved by following similar presentation of Lewis (2003). Specifically, please mention that your X(P) is the same as f(p) in Lewis (2003) and point it out that it is ad hoc. Please also point out that the expression of translation rate (DP/dt) and transcription rate (dM/dt) in Fig. 1. Please carefully provide the details of the references of the parameters in Table 1. Reference [9] (Lewis 2003) does not contain parameter table but includes a list some of them in the text with some references.
Author Response
Reviewer 2
- The model was presented with minimum details and justifications. This can be improved by following similar presentation of Lewis (2003). Specifically, please mention that your X(P) is the same as f(p) in Lewis (2003) Further text has been added in the model derivation.
>>>>We note that in this study X(t) represents the concentration of an intermediate molecule that is under the same transcriptional control as Hes gene but also inhibits translation. Conceptually, it is not the same as f(p) in the Lewis model which represents the dependence of transcription on protein dimers. However, because of the assumptions made about X, the functional form ends up being proportional to the transcription rate f(p). At the point where we introduce the transcriptional regulation using the functional form proposed in the Lewis paper, we now add a citation.
- ... point it out that it is ad hoc.
>>>>It is well established that Hes dimers inhibit Hes transcription. Assuming that dimerisation occurs on a faster time scale than other processes one can derive the functional form for transcriptional repression.
- Please also point out that the expression of translation rate (DP/dt) and transcription rate (dM/dt) in Fig. 1.
>>>>dP/dt represents the total rate of change of protein with respect to time, i.e.production (translation of new protein) + degradation. Similarly, dM/dt 2 represents the total rate of change of mRNA with respect to time, i.e. production (transcription) + degradation. In Figure 1 (b) we plot only the production rates for M and P. Hence it is not appropriate to label these as suggested.
- Please carefully provide the details of the references of the parameters in Table 1. Reference [9] (Lewis 2003) does not contain parameter table but includes a list some of them in the text with some references.
>>>>We have not used precise values from the Lewis paper as the models cannot be directely mapped on to one another. We have used the basal transcription rate and protein degradation rate to be of the same order as those considered by Lewis. However, the model is sensitive to the ratio of k4 to k5.
In a follow up paper we propose to comprehensively analyse the behaviour of the model in different regions of parameter space. However, this is quite a lengthy exercise and we suggest out of scope for this study.